# Zbtb14 regulates monocyte and macrophage development through inhibiting *pu.1* expression in zebrafish

**Yun Deng[1,2], Haihong Wang[1,2], Xiaohui Liu[1,2], Hao Yuan[1,2], Jin Xu[3], Hugues de Thé[2,4], Jun Zhou[1,2]\*, Jun Zhu[2,4]\***

[1]Shanghai Institute of Hematology, State Key Laboratory of Medical Genomics, National Research Center for Translational Medicine at Shanghai, Ruijin Hospital, Shanghai Jiao Tong University School of Medicine, Shanghai, China; [2]CNRS-LIA Hematology and Cancer, Sino-French Research Center for Life Sciences and Genomics, Ruijin Hospital, Shanghai Jiao Tong University School of Medicine, Shanghai, China; [3]Laboratory of Immunology and Regeneration, School of Medicine, South China University of Technology, Guangzhou, China; [4]Université de Paris 7/ INSERM/CNRS UMR 944/7212, Equipe Labellisée Ligue Nationale Contre le Cancer, Hôpital St. Louis, Paris, France

**Abstract** Macrophages and their precursor cells, monocytes, are the first line of defense of the body against foreign pathogens and tissue damage. Although the origins of macrophages are diverse, some common transcription factors (such as PU.1) are required to ensure proper development of monocytes/macrophages. Here, we report that the deficiency of *zbtb14*, a transcription repressor gene belonging to *ZBTB* family, leads to an aberrant expansion of monocyte/macrophage population in zebrafish. Mechanistically, Zbtb14 functions as a negative regulator of *pu.1*, and SUMOylation on a conserved lysine is essential for the repression activity of Zbtb14. Moreover, a serine to phenylalanine mutation found in an acute myeloid leukemia (AML) patient could target ZBTB14 protein to autophagic degradation. Hence, *ZBTB14* is a newly identified gene implicated in both normal and malignant myelopoiesis.

**\*For correspondence:**
zj10802@rjh.com.cn (JZ);
zhuj1966@yahoo.com (JZ)

**Competing interest:** The authors declare that no competing interests exist.

## Editor's evaluation

This manuscript by Deng et al. is a valuable evaluation of zbtb14 and its role in normal myelopoiesis. The authors provided convincing data supporting the role played by zbtb14 in monocyte and macrophage development and its regulation involving the modulation of PU.1 expression. The finding that a mutation in ZBTB14 exists in AML patients also implies how important this gene product is in normal human myelopoiesis.

## Introduction

Hematopoiesis is the process by which hematopoietic stem cells (HSCs) proliferate and differentiate into all blood lineages. It is driven by a variety of transcription factors, which function in a stage and lineage-specific manner (*Bodine, 2017*). A major goal to explore these transcription factors and their regulatory networks is to gain an intensive insight into normal hematopoiesis and its malignant counterpart, leukemia.

Macrophages are key players in many biological processes such as immune response to foreign pathogens and tissue homeostasis. The developmental origin of macrophages is diverse (*Kurotaki*

*et al., 2017*). Most tissue-resident macrophages arise from the blood islands of the yolk sac (*Kurotaki et al., 2017*). Yet, monocytes in circulation are derived from HSCs, which give rise to monocytes in a step-wise manner via common myeloid progenitors, granulocyte-monocyte progenitors, monocyte-dendritic cell progenitors, and common monocyte progenitors (*Hettinger et al., 2013*). During infection, circulating monocytes migrate into tissues and generate inflammatory macrophages (*Menezes et al., 2016*). Whatever the origin, some common transcription regulators (such as PU.1) (*Glass and Natoli, 2016*; *Kueh et al., 2013*) and signaling pathways (such as macrophage colony-stimulating factor receptor [M-CSFR]) are required in monocyte and macrophage development (*Dai et al., 2002*).

A total of 49 zinc finger and BTB domain (ZBTB) containing transcription factors have been identified in human genome. The C-terminal zinc finger motifs of ZBTB proteins enable the binding with DNA, whereas the N-terminal BTB motif mediates the homo/hetero-dimerization/multimerization between different ZBTB family members and recruits corepressors such as NCoR and SMRT (*Maeda, 2016*). Thus, ZBTBs mostly function as transcription repressors.

Several members of ZBTB family transcription factors including ZBTB16 (also known as promyelocytic leukemia zinc finger) (*Sobas et al., 2020*), ZBTB27 (also known as B cell leukemia/lymphoma 6) (*Nakamura, 2000*), ZBTB7 (also known as leukemia/lymphoma-related factor) (*Constantinou et al., 2019*), and ZBTB15 (also known as T-helper-inducing POZ/Kruppel-like factor) (*Taniuchi, 2016*) play various roles in both normal and malignant hematopoiesis in humans. In addition, at least 12 *Zbtb* genes are involved in hematopoietic development in mice (*Maeda, 2016*).

ZBTB14, a ZBTB family member whose function has been poorly characterized, is expressed in a variety of blood cell types (The Human Protein Atlas). Recently, a missense mutation of *ZBTB14* gene (*ZBTB14^{S8F}*) was detected by whole-exome sequencing in a newly diagnosed acute myeloid leukemia (AML) patient (*Tyner et al., 2018*). Note that mutation of *ZBTB14* has not previously been identified in AML. Nevertheless, the potential role of *ZBTB14* in hematopoiesis and leukemogenesis was obscure, (*Yin et al., 2015*).

In the present work, we provide in vivo evidence showing that the deficiency of *zbtb14* leads to an expansion of monocyte/macrophage population in zebrafish. Mechanistic studies reveal that Zbtb14 functions as a negative regulator of *pu.1*, and SUMOylation on a conserved lysine is essential for the transcriptional repression of Zbtb14. In addition, human ZBTB14^{S8F} mutant protein is demonstrated as a loss-of-function transcription factor. Hence, our results for the first time not only unravel the physiological function of Zbtb14 during monocyte/macrophage development, but also elucidate the defective role of its mutant in AML.

## Results

### Generation of a *zbtb14*-deficient zebrafish line

The zebrafish serves as an ideal model organism for hematopoietic development and disease studies (*Gore et al., 2018*). Zebrafish Zbtb14 protein shares 70% homology to its human counterpart ZBTB14. The two transcription factors bear a nearly identical N-terminal BTB motif and five consecutive C-terminal zinc finger motifs (*Figure 1A*).

To explore the roles of *zbtb14* in hematopoiesis, especially in myelopoiesis, a mutant zebrafish line was established by using the CRISPR/Cas9 system targeting the BTB domain of Zbtb14. Ten nucleotides were deleted, which resulted in a truncated protein with only 80 amino acids by frameshifting (*Figure 1B and C*). The mutant *zbtb14* gene was expressed in HEK293T cells, and western blot analysis revealed a short protein as anticipated (*Figure 1D*). Note that the presence of a weak adduct, implying certain covalent modification of Zbtb14 protein might exist.

### Deficiency of *zbtb14* specifically affects monocyte and macrophage development

Like mammalian hematopoiesis, zebrafish hematopoiesis consists of primitive and definitive waves which occur sequentially in distinct anatomical sites (*Galloway and Zon, 2003*; *Zon, 1995*). The primitive hematopoiesis gives rise to embryonic myeloid cells (neutrophils and macrophages) and erythrocytes from two intraembryonic locations, the rostral blood island (the equivalent of mammalian yolk sac) and intermediate cell mass. The definitive HSCs which give rise to all blood lineages initiate within the ventral wall of the dorsal aorta (a tissue analogous to the mammalian aorta/gonad/mesonephros),

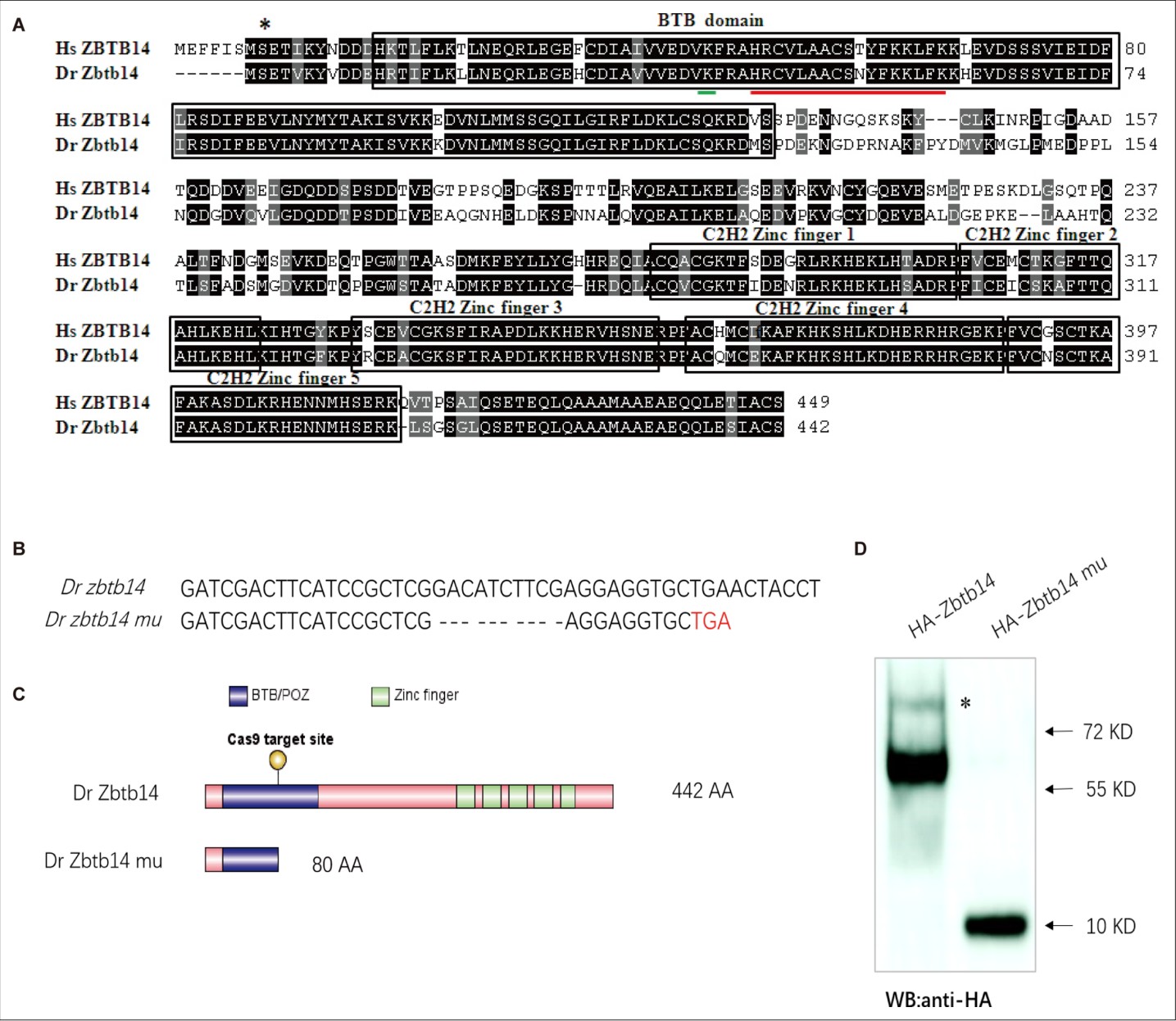

**Figure 1.** The establishment of a zebrafish *zbtb14* knockout line. (**A**) Sequence alignment of. ZBTB14 and Zbtb14 proteins. Hs: *Homo sapiens*, Dr: *Danio rerio*. The conserved BTB domain and C2H2 zinc finger domains are boxed. *: The mutated serine identified in an acute myeloid leukemia (AML) patient. The putative SUMOylated lysine K40 and the nuclear localization signal (NLS) are underlined, respectively. (**B**) Schematic representation of Cas9 target site in the first exon of zebrafish *zbtb14*. The deleted nucleotides in the mutant gene are marked by hyphens. (**C**) Schematic representation of wild type (442 amino acids) and mutant Zbtb14 proteins (80 amino acids). (**D**) Western blot analysis of HA-tagged wild type and mutant Zbtb14 proteins expressed in HEK293 cells. * indicates the adduct band.

The online version of this article includes the following source data for figure 1:

**Source data 1.** Source data for *Figure 1D*.

then translocate to the caudal hematopoietic tissue (the equivalent of fetal liver) and colonize in kidney marrow (the equivalent of bone marrow) in adults (*Bertrand et al., 2008*; *Bertrand et al., 2007*).

To unravel the role of *zbtb14* during embryonic hematopoiesis, whole-mount in situ hybridization (WISH) analyses were performed with multiple hematopoietic lineage-specific markers in *zbtb14*-deficient embryos and larvae. A significant increase of macrophage markers including *mfap4*, *csf1ra*,

and *mpeg1.1* (*Meijer et al., 2008*; *Spilsbury et al., 1995*; *Zakrzewska et al., 2010*) was observed from 19.5 hr post-fertilization (hpf) to 3 days post-fertilization (dpf) (*Figure 2A–G' and H*). This expanded macrophage population was further confirmed in *zbtb14$^{-/-}$//Tg(mpeg1.1:eGFP)* larvae (*Figure 2I–J' and K*). It is worth noting that the *zbtb14*-deficient macrophages could still migrate toward the wound as wild type ones did, implying their function remains intact (*Figure 2L–M'*). In addition, WISH with *apoeb* probe (a specific microglia marker) and neutral red vital dye staining revealed an expansion of microglia (the macrophages that reside in brain) in *zbtb14*-deficient larvae (*Figure 2N–Q' and R*).

The primitive macrophages cannot sustain for a long period, which will eventually be replaced with HSC-derived macrophages as definitive hematopoiesis begins (*Xu et al., 2012*). Since *zbtb14* mutant zebrafish were viable and fertile, the whole kidney marrow (WKM) samples were collected from wild type *Tg(mpeg1.1:eGFP)* and *zbtb14$^{-/-}$//Tg(mpeg1.1:eGFP)* lines in adults. The myeloid cell fraction was analyzed, and many more macrophages were found in *zbtb14$^{-/-}$//Tg(mpeg1.1:eGFP)* zebrafish than in controls (1.82% *mpeg1.1$^+$* versus 0.50% *mpeg1.1$^+$*) (*Figure 2—figure supplement 1S–T' and U*), indicating that the development of macrophages originated from HSCs are also affected.

In addition, the expressions of neutrophil and erythrocyte markers during primitive hematopoiesis stage, as well as those of hematopoietic stem and progenitor cells, neutrophil, erythrocyte, and lymphocyte markers during definitive hematopoiesis stage were all normal (*Figure 2—figure supplement 1*). These observations suggest that only the development of monocyte/macrophage lineage is affected in the absence of *zbtb14*.

## Overproliferation of monocyte/macrophage progenitor is the main cause of abnormal macrophage lineage expansion in *zbtb14* mutants

The aberrant expanded macrophage population in *zbtb14*-defective mutants could be caused by either increased proliferation or decreased apoptosis rate. To distinguish between the two possibilities, antiphosphohistone H3 (pH3) immunostaining and terminal deoxynucleotidyl transferase dUTP nick end labeling (TUNEL) assays were carried out to evaluate the proliferation and apoptosis status of macrophages, respectively.

A significant increase of pH3$^+$GFP$^+$ double-positive cells were observed in *zbtb14$^{-/-}$//Tg(mpeg1.1:eGFP)* larvae compared to those in controls (*Figure 3A–C' and D*). Nevertheless, the results from TUNEL assay could not show discernible differences in the percentage of TUNEL$^+$GFP$^+$ cells in *zbtb14$^{-/-}$//Tg(mpeg1.1:eGFP)* larvae compared to the percentage in controls (*Figure 3E–G' and H*). These data suggest that the expanded macrophage population is due to enhanced proliferation.

Moreover, the *mpeg1.1$^+$* cells were isolated from the myeloid fraction of WKM in adult zebrafish (10-month-old) and subjected to May-Grünwald-Giemsa (MGG) staining. According to their morphology, the cells were classified into three populations: proliferative immature progenitors (I), intermediate (II), and non-proliferative mature cells (III) (*Figure 3I*). The results revealed that the proportion of immature progenitors (with a round-shape morphology and a higher nuclear/cytoplasm ratio) was obviously increased in *zbtb14* mutants compared with the siblings (*Figure 3J*). By contrast, the proportion of fully differentiated macrophages was decreased in the mutants (*Figure 3J*).

Based on these observations, we can draw the conclusion that the expansion of macrophage lineage is mainly due to excessive monocyte/macrophage progenitor proliferation in *zbtb14* mutants.

## Zbtb14 functions as a transcription repressor of *pu.1* in regulating monocyte and macrophage development

The monocyte/macrophage abnormalities in *zbtb14*-deficient larvae could be effectively rescued with either zebrafish *zbtb14* or human *ZBTB14* mRNA, confirming the specificity and conservation of ZBTB14 proteins (*Figure 4A–D and F*).

ZBTB family proteins are frequently described as transcription repressors, nevertheless, ZBTB14 displays an activation or inhibition effect on different promoters (*Kaplan and Calame, 1997*). To distinguish the activity of Zbtb14 on transcription, the repression domain of DAXX (a transcription corepressor) (*Zhou et al., 2006*) was fused in frame with Zbtb14 (DAXX-Zbtb14), which forced the fusion protein to be a potent repressor. In vivo rescue assays demonstrated that *DAXX-zbtb14* mRNA had an obvious rescue effect as wild type *zbtb14* mRNA (*Figure 4E and F*), implying Zbtb14 acted as a negative regulator in monocyte/macrophage development.

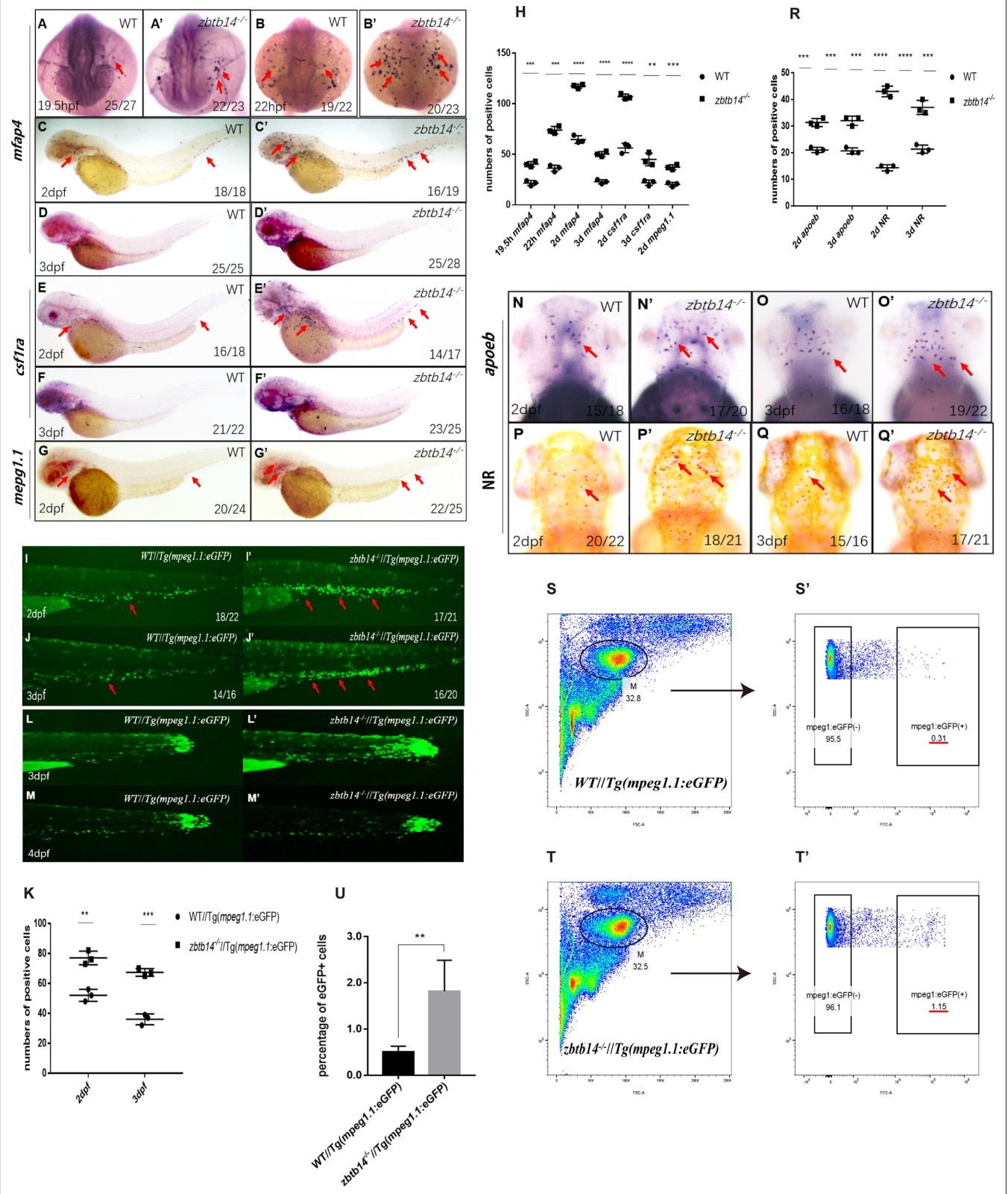

**Figure 2.** Deficiency of *zbtb14* specifically impairs monocyte and macrophage development in embryonic and adult zebrafish. (**A–G'**) Whole-mount in situ hybridization (WISH) analyses of macrophage markers *mfap4* (**A–D'**), *csf1ra* (**E–F'**), *mpeg1.1* (**G, G'**) from 19.5 hr post-fertilization (hpf) to 3 days post-fertilization (dpf) in wild type (WT) and *zbtb14*-deficient embryos and larvae. Red arrows indicate the main positions of positive cells for each marker. n/n, number of embryos/larvae showing representative phenotype/total number of embryos/larvae examined. (**H**) Statistical results for A–G' (Student's

*Figure 2 continued on next page*

*Figure 2 continued*

t test, N=3, 14–28 embryos were used for each probe. Each dot represents the mean value of one experiment, which was obtained from the counts of all of the embryos/larvae in the same group. Error bars represent mean ± standard error of the mean (SEM). **p<0.01, ***p<0.001, ****p<0.0001). (I–J') GFP positive cells were increased in *zbtb14-/-//Tg(mpeg1.1:eGFP)* embryos at 2 and 3 dpf. (K) Statistical results for I–J' (Student's t test, N=3, 14–22 larvae were used for each experiment. Each dot represents the mean value of one experiment. Error bars represent mean ± SEM. **p<0.01, ***p<0.001). (L–M') GFP positive cells in both *Tg(mpeg1.1:eGFP)* and *zbtb14-/-//Tg(mpeg1.1:eGFP)* larvae can migrate to the wound. (N–Q') *apoeb* and neutral red positive cells were both increased in *zbtb14*-deficient larvae at 2 and 3 dpf. (R) Statistical results for N–Q' (Student's t test, N=3, 15–22 larvae were used for each experiment. Each dot represents the mean value of one experiment. Error bars represent mean ± SEM. ***p<0.001, ****p<0.0001). (S–T') Representative scatterplot generated by FACS analysis of WKM samples collected from WT *Tg(mpeg1.1:eGFP)* (up panel) and *zbtb14-/-//Tg(mpeg1.1:eGFP)* (bottom panel) zebrafish lines in 4-month-old adults. M: myeloid gate. (U) Statistical results for S–T' in WT *Tg(mpeg1.1:eGFP)* and *zbtb14-/-//Tg(mpeg1.1:eGFP)* zebrafish. (Student's t test, N=4, each time one male and one female were used in the WT and mutant groups. Error bars represent mean ± SEM. **p<0.01).

The online version of this article includes the following figure supplement(s) for figure 2:

**Figure supplement 1.** Expression of lineage-specific markers during primitive and definitive hematopoiesis stages in *zbtb14*-deficient embryos.

To elucidate the mechanism underlying the aberrant monocyte/macrophage development, we performed RNA sequencing (RNA-seq) analyses on *mpeg1.1+* cells isolated from *Tg(mpeg1.1:eGFP)* and *zbtb14-/-//Tg(mpeg1.1:eGFP)* larvae at 2 dpf. The differentially expressed genes (DEGs) analysis of the RNA-seq data indicated that multiple important regulators involving monocyte/macrophage development including *pu.1*, *csf1ra*, *csf1b*, *il34*, *mafb*, *klf4*, *irf8*, and *c-myc* were upregulated (*Figure 4—figure supplement 1*). The expression level of *pu.1*, the pivotal transcription factor which can promote macrophage proliferation (*Dai et al., 2002*), was obviously increased. Such upregulation was further confirmed by real-time quantitative PCR (RT-qPCR) analyses (*Figure 4G*). In addition, WISH analyses showed that the signals of *pu.1* were clearly intensified from 18 to 22 hpf in *zbtb14*-deficient embryos compared with the wild type ones (*Figure 4H–K' and L*).

Since Zbtb14 was identified as a transcription repressor, we postulated that *pu.1* would be a major direct target of Zbtb14, whose derepression in the absence of Zbtb14 probably contributed to the expansion of monocyte/macrophage population. To test this hypothesis, a –0.6 kb zebrafish *pu.1* promoter in luciferase reporter was co-transfected with *zbtb14* expressing plasmid in HEK293T cells. As anticipated, luciferase analyses showed that Zbtb14 displayed a significant repression effect (*Figure 4M*).

Next, in vivo chromatin immunoprecipitation polymerase chain reaction (ChIP-PCR) was conducted in zebrafish larvae expressing GFP or GFP-Zbtb14 using an anti-GFP antibody. In this experiment, the *pu.1* promoter region could be specifically co-immunoprecipitated with GFP-Zbtb14 (*Figure 4N*).

To further demonstrate that *pu.1* was upregulated in macrophage lineage, a series of in vivo experiments was performed. A prominent rescue effect could be obtained with a dominant-negative *Pu.1* plasmid (the DBD domain of Pu.1 was under the control of *mpeg1.1* gene's promoter, cloned in TOL2 backbone) injection in *zbtb14* mutants (*Figure 4O and F*). Moreover, we took advantage of a zebrafish *pu.1^Δ371^* mutant line (a truncated Pu.1 whose transactivation activity was reduced) in which macrophages were reduced (*Yu et al., 2017*; *Figure 4P and F*), and no obvious alleviation could be found in *zbtb14-/-//pu.1^Δ371^* double mutant zebrafish, indicating *zbtb14* was epistatic to *pu.1* (*Figure 4Q and F*).

In summary, these findings suggest that Zbtb14 regulates monocyte and macrophage development through inhibiting the expression of *pu.1*.

## SUMOylation is essential for the transcription repression of Zbtb14

Post-translational modification plays important roles in regulating the functions of substrate proteins. Similar to ubiquitination, protein SUMOylation is catalyzed by a sequential enzymatic cascade including E1 (SAE1/SAE2), E2 (UBC9), and E3 in which SUMO (Small Ubiquitin-like MOdifier) molecules are covalently attached to lysine residues within substrate proteins (*Eifler and Vertegaal, 2015*). The SUMOylation is generally associated with transcriptional repression through the recruitment of corepressors such as NCoR and SMRT (*Garcia-Dominguez and Reyes, 2009*; *Valin and Gill, 2007*). We have mentioned above the presence of a weak adduct which was ~10 kD (the size of one SUMO molecule) larger than the unmodified Zbtb14 protein (*Figure 1D*). Combing the fact that Zbtb14 was a repressor, we reasoned Zbtb14 would be a SUMOylated substrate.

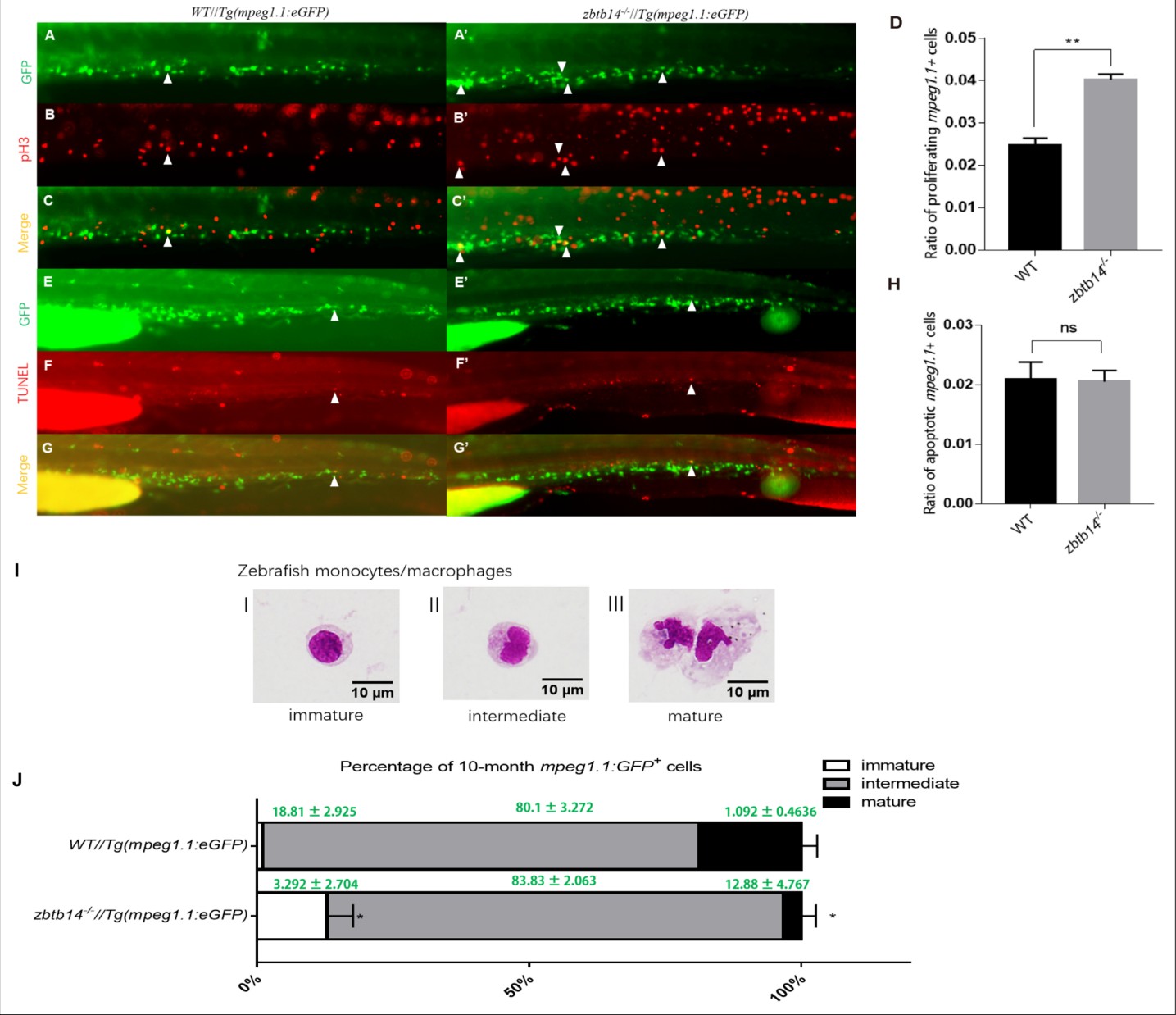

**Figure 3.** Overproliferation of macrophage progenitor is the main cause of macrophage lineage expansion in *zbtb14* mutants. (**A–C', E–G'**) pH3 and transferase dUTP nick end labeling (TUNEL) assays in *zbtb14⁻/⁻*//Tg(*mpeg1.1*:eGFP) and Tg(*mpeg1*:EGFP) control larvae. Triangles indicate positive signals. (**D, H**) Statistical result for A–C' and E–G'. Data shown are the means ± standard error of the mean (SEM) of at least 15 larvae. ns: not statistically significant, **p<0.01. (**I**) Classification of zebrafish monocytes/macrophages is based on their morphology. (**J**) Quantification of *mpeg1.1⁺* cells at each differentiation stage. Sorted macrophage lineage cells from 10-month-old *zbtb14⁻/⁻*//Tg(*mpeg1.1*:eGFP) and Tg(*mpeg1.1*:eGFP) larvae were subjected to May-Grünwald-Giemsa staining and separated into immature, intermediate, and mature groups according to their morphology. Scale bar: 10 µm. Error bars represent mean ± SEM of three independent experiments. *p<0.05 (Student's t-test).

To validate this hypothesis, *zbtb14*-expressing plasmid was transfected with or without *UBC9* and *SUMO1* in HEK293T cells. Western blot analyses showed that the adduct band became much more intensive in the presence of UBC9 and SUMO1 (*Figure 5A*, lanes 1 and 2). Furthermore, immuno-coprecipitation assays showed that SUMO1 molecules could be coprecipitated with Zbtb14 (*Figure 5B*, lane 1). These results indicated that Zbtb14 could be SUMOylated in cells.

SUMOylation is a process by which the SUMO monomer/polymer is covalently ligated to specific lysine residues of the target protein (*Chang and Yeh, 2020*). Dozens of potential SUMOylation sites which spread throughout the protein were predicted by bioinformatics (SUMOsp2.0 prediction

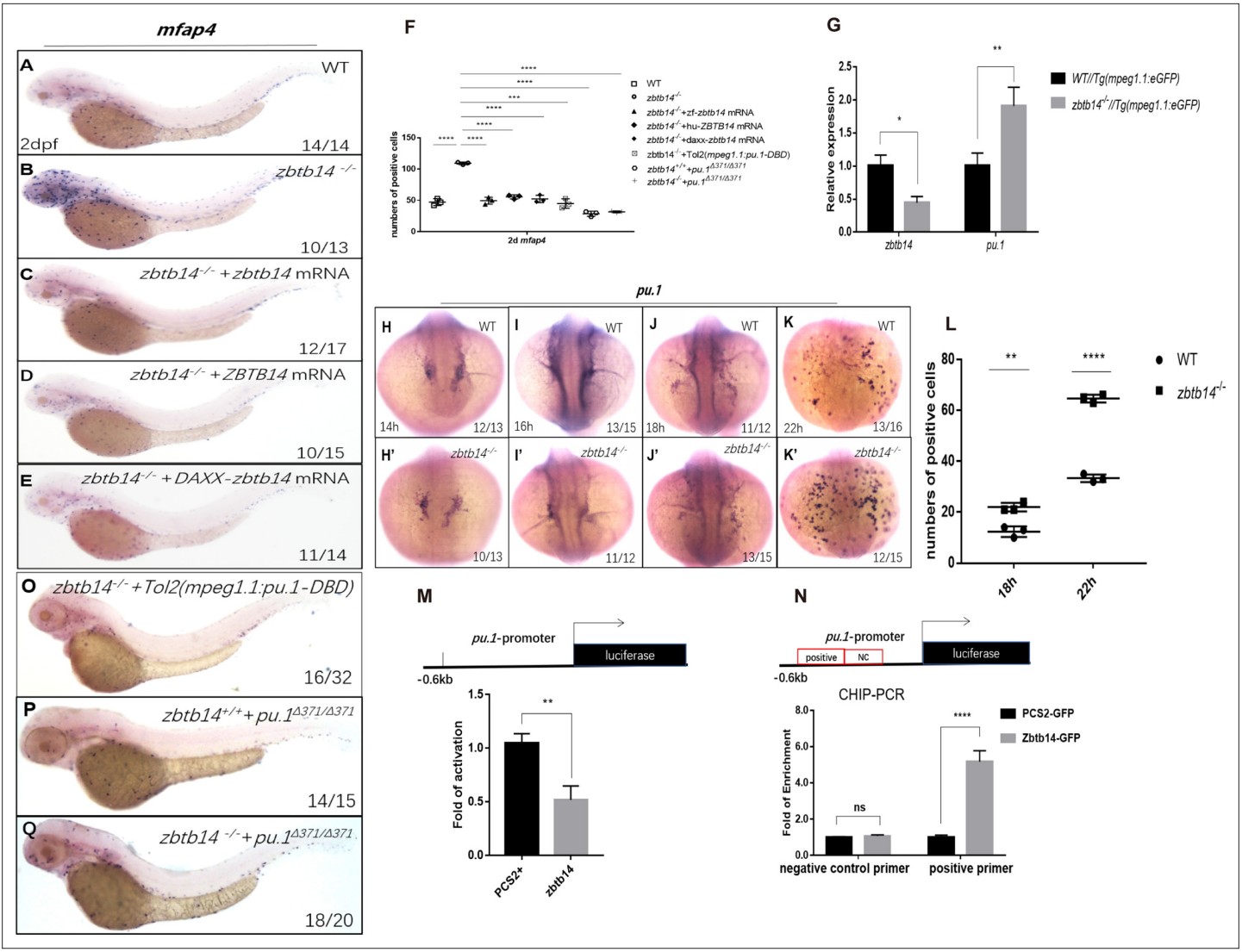

**Figure 4.** Zbtb14 regulates monocyte and macrophage development through inhibiting the expression of *pu.1*. (**A–E**) mRNA rescue assays in *zbtb14*-/- larvae. *mfap4* probe was used in whole-mount in situ hybridization (WISH) to examine rescue effect with wild *zbtb14* (**C**), *ZBTB14* (**D**), *DAXX-ZBTB14* (**E**) mRNA injections. (**F**) Statistical result for A–E and O–Q. The statistical significance was calculated by using one-way analysis of variance (ANOVA). The statistical significance was calculated using one-way ANOVA followed by Dunnett T3 correction. The asterisk indicates a statistical difference. (N=3, 10–32 larvae were used for each experiment. Each dot represents the mean value of one experiment. Error bars represent mean ± standard error of the mean (SEM). ****p<0.0001.) (**G**) Quantitative reverse transcriptase polymerase chain reaction analysis of *zbtb14* and *pu.1* in GFP positive cells enriched from *Tg(mpeg1.1:eGFP)* and *zbtb14*-/-//*Tg(mpx:eGFP)* larvae at 2 days post-fertilization (dpf). To determine the relative expression rate, data were normalized to the expression level of wild type (WT) groups (which were set to 1.0) after normalized to the internal control of *β-actin* (Student's t test, N=3. Error bars represent mean ± SEM. *p<0.05, **p<0.01). (**H–K'**) Serial WISH analyses of *pu.1* in WT and *zbtb14*-/- embryos. (**L**) Statistical results for H–K' (Student's t test, N=3, 10–16 embryos were used. Each dot represents the mean value of one experiment, which was obtained from the counts of all of the embryos in the same group. Error bars represent mean ± SEM. **p<0.01, ****p<0.0001.) (**M**) Luciferase reporter assay of Zbtb14 on the *pu.1* promoter. Bars showed the relative luciferase activity on the zebrafish *pu.1* promoter (–0.6 kb). (Student's t test, N=3. Error bars represent mean ± SEM. **p<0.01.) (**N**) Chromatin immunoprecipitation polymerase chain reaction (ChIP-PCR) analysis of *pu.1* promoter in zebrafish larvae expressing GFP or Zbtb14-GFP using an anti-GFP antibody. Positive: the location of the positive primers. NC: the location of the negative control primers. The statistical significance was calculated by using one-way ANOVA. The asterisk indicates a statistical difference. (N=3. Error bars represent mean ± SEM. ns: not statistically significant, ****p<0.0001.) (**O–Q**) WISH assay of *mfap4* in *zbtb14*-/- mutants injected with TOL2 *mpeg1.1:Pu.1 DBD*, *pu.1^{Δ371/Δ371}* mutants, and *zbtb14*-/-//*pu.1^{Δ371/Δ371}* double mutants.

The online version of this article includes the following figure supplement(s) for figure 4:

**Figure supplement 1.** Heat map of genes associated with monocyte/macrophage development in *mpeg1.1+* cells isolated from *Tg(mpeg1.1:eGFP)* and *zbtb14*-/-//*Tg(mpeg1.1:eGFP)* larvae at 2 days post-fertilization (dpf).

*Figure 4 continued on next page*

**Figure supplement 2.** The role of *zbtb14* in macrophage proliferation is lineage specific.

**Figure supplement 3.** Expression of *zbtb14* transcript in *mpx*[+] and *mpeg1.1*[+] cells at different developmental stages.

**Figure supplement 4.** Luciferase reporter assay of Pu.1 on the *zbtb14* promoter.

**Figure supplement 5.** Expression of *c-myc* transcript in *mpeg1.1*[+] cells.

software, K259 has the highest score among 41 predicted SUMOylated sites). We carried out a series of mutations and finally found that the adduct band of Zbtb14$^{K40R}$ mutant was abolished (*Figure 5A*, lanes 3–6), suggesting K40 was the SUMOylated site.

Since K40 is located adjacent to the bipartite nuclear localization sequence of Zbtb14 (*Sugiura et al., 1997*; *Figure 1A*), we questioned whether SUMOylation would affect its protein subcellular localization. HA-tagged Zbtb14 and Zbtb14$^{K40R}$ were transfected in HEK293T cells, respectively. The results from immunofluorescence analyses revealed that the mutant protein was located in the nucleus as wild type Zbtb14 (*Figure 5C*).

Next, luciferase reporter assays with the –0.6 kb zebrafish *pu.1* promoter were carried out. While Zbtb14 and SUMO1-Zbtb14$^{K40R}$ (SUMO1 molecule was fused in frame with Zbtb14$^{K40R}$ to mimic the SUMOylated Zbtb14) displayed a strong repression effect, Zbtb14$^{K40R}$ completely lost the ability to repress transcription (*Figure 5D*).

Finally, in vivo rescue assays were conducted in *zbtb14*-deficient larvae with *zbtb14*, *zbtb14$^{K40R}$*, and *SUMO1-zbtb14$^{K40R}$* mRNA, respectively. As expected, both *zbtb14* and *SUMO1-zbtb14$^{K40R}$* mRNAs had a remarkable rescue effect, whereas *zbtb14$^{K40R}$* mRNA did not (*Figure 5E and F*).

Taken together, these results suggest that SUMOylation of Zbtb14 is pivotal for transcription repression.

## Human ZBTB14$^{S8F}$ mutant is a loss-of-function transcription factor

A missense mutation, *ZBTB14$^{S8F}$*, was detected in a de novo AML patient (*Tyner et al., 2018*). To assess the role of the mutant protein, we first performed in vivo rescue assays in *zbtb14*-deficent zebrafish. While the wild type *ZBTB14* mRNA significantly rescued the expanded macrophage population, the mutant failed to display any rescue effect (*Figure 6A and B*), implying that the normal functions of ZBTB14 was lost.

Next, we investigated the reason underlying the defects of the ZBTB$^{S8F}$ mutant. Although its transcription level was comparable to that of wild type *ZBTB14* (*Figure 6C*), ZBTB14$^{S8F}$ protein could hardly be detected by western blot analysis, implying that the stability of the mutant protein was impaired (*Figure 6D*, lane 4). We thus treated *ZBTB14$^{S8F}$*-expressing HEK293T cells with MG132 (a proteasomal inhibitor), or bafilomycin (an autophagy inhibitor). The results indicated that bafilomycin, but not MG132, inhibited ZBTB14$^{S8F}$ protein turnover (*Figure 6D*, lanes 7 and 10), suggesting that the mutant protein might be subjected to autophagic degradation. Since serine is most frequently modified with phosphorylation, which could affect the protein stability, ZBTB14$^{S8A}$ and ZBTB14$^{S8D}$ mutants were constructed to mimic the dephosphorylated or constitutively phosphorylated status of ZBTB14. However, the turnover of both two mutant proteins still kept normal (*Figure 6D*, lanes 2 and 3).

Autophagy is a process in which cellular material is degraded through the lysosomal pathway and recycled. Selective autophagy is mediated by specific cargo receptors which interact with the Atg8 (autophagy-related 8) family proteins and thereby link the cargo with the autophagy machinery (*Fracchiolla et al., 2017*). MAP1LC3/LC3 (microtubule-associated protein1 light chain 3) is one of the most important autophagy-related proteins which participates in autophagosome formation (*Fracchiolla et al., 2017*). LC3 interacts with LIR (LC3-interacting region) motif (also termed as AIM [Atg8-interacting motif]) of selective autophagy receptors that carry cargo for degradation. The LIR/AIM motif contains the consensus sequence [W/F/Y]xx[L/I/V] (*Popelka and Klionsky, 2015*). The aromatic residue (W/F/Y) and hydrophobic residue (L/I/V) bind to two hydrophobic pockets formed by the ubiquitin-like fold of the Atg8 proteins (*Atkinson et al., 2019*). Thus, the LIR/AIM ensures the selectivity of the interaction between Atg8 proteins and their binding partners. We noticed that the S to F mutation (FETI) happened to form a φxxΨ motif, which was probably the reason why the ZBTB14$^{S8F}$ mutant protein was targeted to undergo autophagic degradation. To further validate this point, ZBTB14$^{S8W}$ and ZBTB14$^{S8Y}$ mutants were also constructed and expressed in HEK293T cells. The results

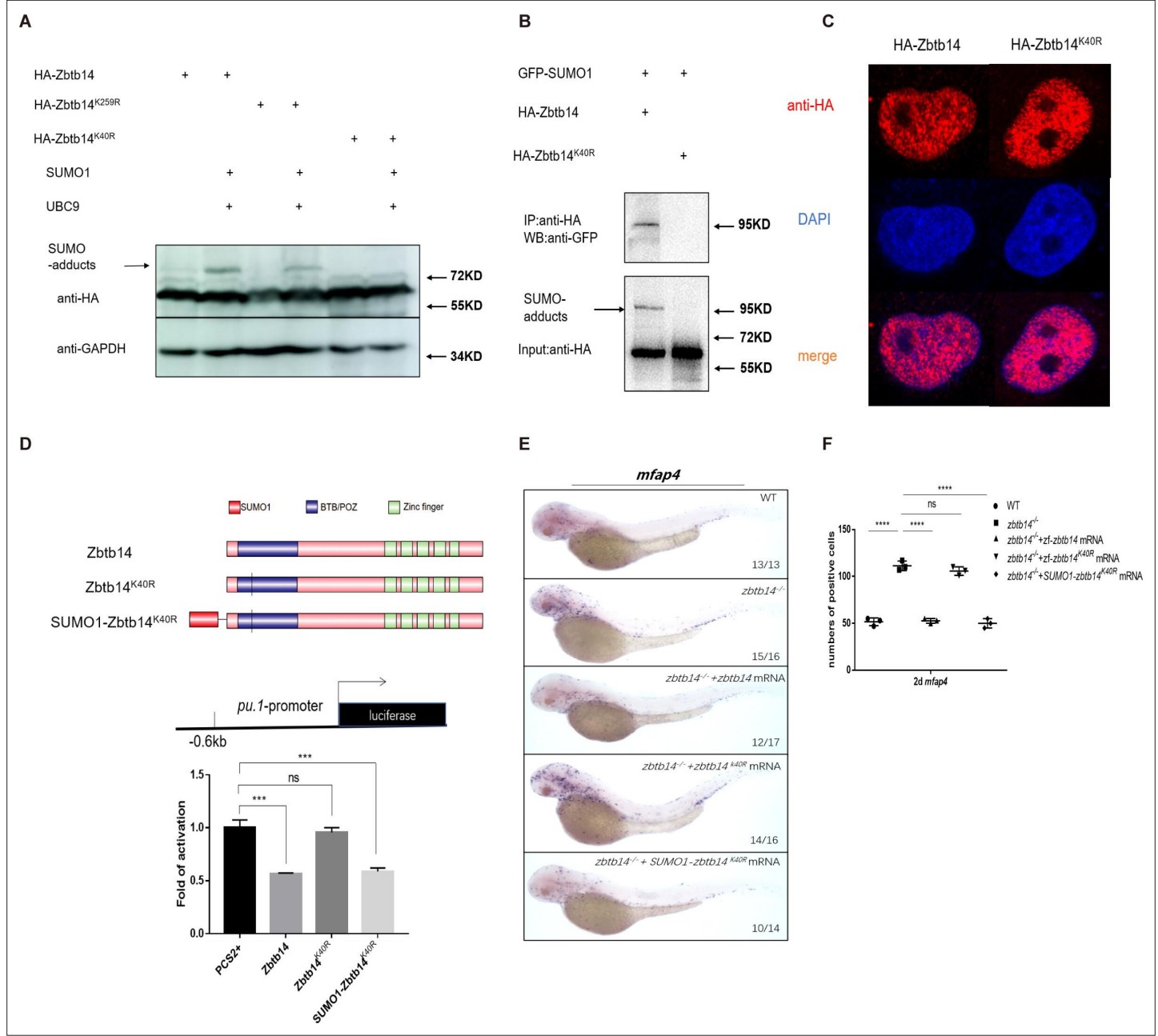

**Figure 5.** SUMOylation is indispensable for transcription repression of Zbtb14. (**A**) Western blot analysis (anti-HA) of HA-tagged wild type (WT), Zbtb14^K40R, and Zbtb14^K259R mutant proteins expressed in HEK293T cells in the absence or presence with the SUMO (Small Ubiquitin-like MOdifier) conjugating enzyme UBC9 and SUMO1. (**B**) HA-tagged WT or Zbtb14^K40R mutant protein was immunoprecipitated with an anti-HA antibody from HEK293T cells co-expressing GFP-SUMO1, and SUMOylated Zbtb14 protein was detected by western blot with an anti-GFP antibody. (**C**) Immunofluorescence analysis of WT (left panel) and Zbtb14^K40R mutant protein (right panel). (**D**) The structure of variant forms of Zbtb14, including WT, Zbtb14^K40R, and SUMO1-Zbtb14^K40R mutants (top panel). Repression of luciferase expression from the zebrafish *pu.1* promoter (–0.6 kb) by Zbtb14 mutants (bottom panel). Bars showed the relative luciferase activity on the zebrafish *pu.1* promoter (–0.6 kb). (Student's t test, N=3. Error bars represent mean ± standard error of the mean (SEM). ns: not statistically significant, ***p<0.001.) (**E**) mRNA rescue assays in *zbtb14^-/-* mutant larvae. *mfap4* probe was used in whole-mount in situ hybridization (WISH) to examine rescue effects of injections of *zbtb14*, *zbtb14^K40R*, and *SUMO1-zbtb14^K40R* mRNA. (**F**) Statistical result for E. The statistical significance was calculated by using one-way analysis of variance (ANOVA). The asterisk indicates a statistical difference. (N=3, 10–17 embryos were used for each experiment. Each dot represents the mean value of one experiment. Error bars represent mean ± SEM. ns: not statistically significant, ****p<0.0001.)

The online version of this article includes the following source data for figure 5:

**Source data 1.** Source data for *Figure 5A*.

**Source data 2.** Source data for *Figure 5B*.

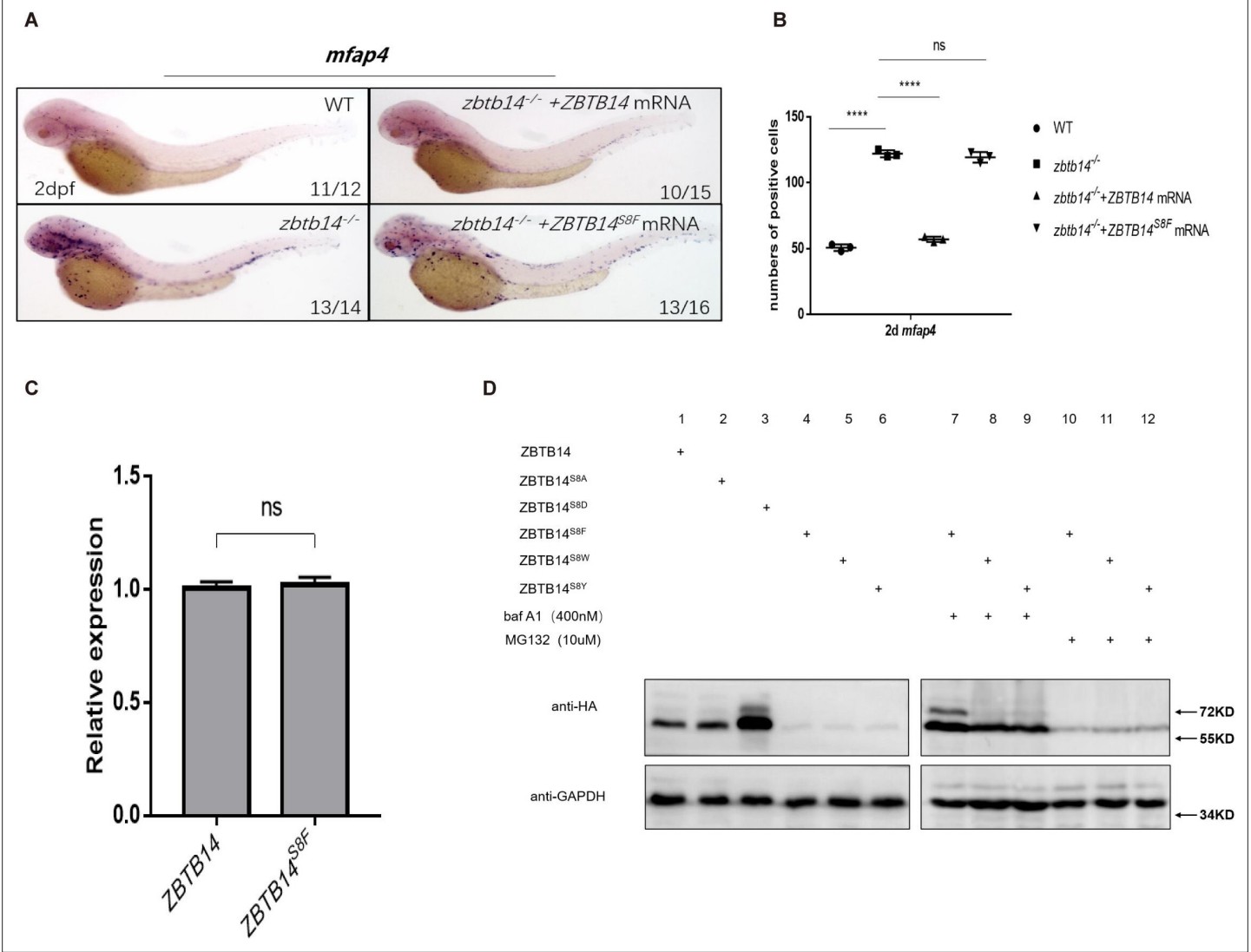

**Figure 6.** Human ZBTB14[S8F] mutant is a loss-of-function transcription factor. (**A**) mRNA rescue assays in *zbtb14*[-/-] mutant larvae. *mfap4* probe was used in whole-mount in situ hybridization (WISH) to examine rescue effects of injections of *ZBTB14* and *ZBTB14*[S8F] mRNA. (**B**) Statistical result for A. The statistical significance was calculated by using one-way analysis of variance (ANOVA). The asterisk indicates a statistical difference. (N=3, 10–16 embryos were used for each experiment. Each dot represents the mean value of one experiment. Error bars represent mean ± standard error of the mean (SEM). ns: not statistically significant, ****p<0.0001.) (**C**) Quantitative reverse transcriptase polymerase chain reaction analysis of *ZBTB14* and *ZBTB14*[S8F] transfected in HEK293T cells. To determine the relative expression rate, data were normalized to the expression level of wild type (WT) groups (which were set to 1.0) after normalized to the internal control of *β-actin*. (Student's t test, N=3. Error bars represent mean ± SEM. ns: not statistically significant.) (**D**) Western blot analysis (anti-HA) of HA-tagged WT, ZBTB14[S8A], ZBTB14[S8D], ZBTB14[S8F], ZBTB14[S8W], and ZBTB14[S8Y] mutant proteins expressed in HEK293T cells in the absence or presence with baf A1 or MG132. GAPDH served as internal control.

The online version of this article includes the following source data for figure 6:

**Source data 1.** Source data for *Figure 6D*.

from western blots showed that the turnover of the two proteins was comparable to that of ZBTB14[S8F] mutant protein (***Figure 6D***, lanes 5, 6, 8, 9, 11, and 12).

Overall, the serine to phenylalanine mutation impairs the protein stability of ZBTB14, which probably contributes to AML pathogenesis.

## Discussion

In this study we characterize the roles of the previously enigmatic Zbtb14 in monocyte/macrophage development. Zbtb14 is absolutely required for proper monopoiesis during primitive hematopoiesis, which is reflected by the expansion of macrophage population in *zbtb14* mutant embryos/larvae compared to wild type ones from 19.5 hpf to 3 dpf. Such a requirement of Zbtb14 continues to definitive hematopoiesis since HSCs-derived macrophages remain hyperproliferative in *zbtb14⁻/⁻* adults. Hence, *zbtb14* is indispensable for the maintenance of proper quantity of monocytes/macrophages. Functionally, the *zbtb14*-deficient *mpeg1.1⁺* cells can still be recruited to the wound, and result in a normal healing, implying these macrophages are able to terminally differentiate into mature cells.

The results from anti-pH3 and MGG staining assays indicate that the abnormal expansion of macrophages is mainly due to excessive monocyte/macrophage progenitor proliferation in *zbtb14* mutants. In addition, DEGs analyses of RNA-seq data further indicate several key regulators involved in macrophage proliferation such as *csf1ra*, *csf1b*, and *il34* (M-CSFR signaling genes) (*Dai et al., 2002*), and *pu.1* (*Celada et al., 1996*) are all significantly upregulated.

The mechanistic studies identify that *pu.1* is a direct target of Zbtb14. As a pivotal ETS family transcription factor, PU.1 is implicated in multiple stages of hematopoiesis such as generation of early myeloid progenitors, cell fate determination of granulocytic versus monocytic lineages of neutrophil-macrophage progenitors, maintenance of the accessibility of macrophage-specific genes during monocytic differentiation (*Kastner and Chan, 2008*). It is worth noting that while the aberrant macrophage expansion could be effectively rescued with dominant-negative Pu.1 (TOL2-*mpeg1.1:pu.1-DBD*), the development of neutrophil lineage still kept normal in *zbtb14* mutants (*Figure 4—figure supplement 2A-A″, B*). In parallel, the overexpression of full-length *pu.1* (TOL2-*mpeg1.1:pu.1*) in wild type embryos could induce an expansion of macrophage population without interfering neutrophil lineage development (*Figure 4—figure supplement 2C-D′, E*). In addition, we compared the expression level of *zbtb14* in *mpeg1.1⁺* and *mpx⁺* cells at different developmental stages, and found *zbtb14* transcripts were constantly higher in the former (*Figure 4—figure supplement 3*). These observations suggest that Zbtb14 is a macrophage lineage-specific transcription factor.

Meanwhile, we also investigated the transcriptional network that involved *zbtb14*. Bioinformatic analyses show that PU.1 is a predicted upstream transcription factor of mammalian *ZBTB14*. The promoter region of zebrafish *zbtb14* was cloned into a reporter vector, and luciferase assay showed that PU.1 displayed a significant repression effect on it (*Figure 4—figure supplement 4*). Therefore, a Zbtb14-Pu.1 negative feedback loop might regulate monocyte and macrophage development.

Approximately 20 genes including *CEBPA*, *RUNX1*, *FLT3*, *DNMT3A*, and *NPM1* are most frequently mutated in AML patients (*Ley et al., 2013*). Nevertheless, some rare mutations can be found in few samples (*Ley et al., 2013*). A missense mutation, *ZBTB14^S8F*, was detected in a de novo AML patient (*Tyner et al., 2018*). In the current study, we took advantage of the *zbtb14*-deficient zebrafish line to demonstrate the functional conservation between the human *ZBTB14* and the zebrafish ortholog *zbtb14*. While wild type *ZBTB14* mRNA displayed a similar rescue effect as *zbtb14*, the *ZBTB14* mutant was shown to be a loss-of-function transcription regulator in monocyte/macrophage development. Furthermore, we demonstrated that the protein stability of ZBTB14^S8F was profoundly affected due to aberrant autophage.

*ZBTB14*, originally named as *ZF5*, was cloned as a transcriptional repressor gene on the murine *c-MYC* promoter (*Yanagidani et al., 2000*). *C-MYC* amplification was identified in some AML patients (*Tang et al., 2021*). Moreover, *c-MYC* is one of the most overexpressed genes in AML (*Handschuh et al., 2018*). Either increased expression or aberrant activation of *c-MYC* plays important roles in leukemogenesis (*Salvatori et al., 2011*). RNA-seq and RT-qPCR experiments showed that *c-myc* was upregulated in our *zbtb14*-deficient macrophages (*Figure 4—figure supplement 5*). In addition, it has been reported that *Znf161* (another alias of *Zbtb14*) knockout mice had a defect in genomic instability, which was associated with higher cancer risk (*Kim et al., 2019*). These results suggest that ZBTB14 would be a tumor suppressor, whose inactivity is tightly related with AML pathogenesis in humans.

# Materials and methods

**Key resources table**

| Reagent type (species) or resource | Designation | Source or reference | Identifiers | Additional information |
|---|---|---|---|---|
| Cell line (*Homo sapiens*) | HEK293T | ATCC | | |
| Antibody | Anti-HA (Rabbit monoclonal) | CST | Cat# 3724 | WB(1:1000) IP(1:50) |
| Antibody | Anti-p-Histone H3 (Ser10)(Rabbit polyclonal) | Santa Cruz | Cat# sc-8656-R | IF(1:200) |
| Antibody | Anti-Digoxigenin-AP Fab fragments(Sheep polyclonal) | Roche | Cat# 11093274910 | WISH(1:5000) |
| Sequence-based reagent | *zbtb14*_F | This paper | qPCR primers | CTCGTGTGTTCGGCAGTAATTG |
| Sequence-based reagent | *zbtb14*_R | This paper | qPCR primers | CTGGAGCGAAATCCTTACTGG |
| Sequence-based reagent | *pu.1*_F | This paper | qPCR primers | TCCCAGCAGTCGTAGTCCTC |
| Sequence-based reagent | *pu.1*_R | This paper | qPCR primers | CCATTTCGCAGAAGGTCAA |
| Sequence-based reagent | *c-myc*_F | This paper | qPCR primers | CAATTCTGGAACGGCATTCG |
| Sequence-based reagent | *c-myc*_R | This paper | qPCR primers | GAAGTAGAAGTAGGGCTGGATG |
| Commercial assay or kit | In Situ Cell Death Detection Kit, TMR red | Roche | Cat# 12156792910 | |
| Commercial assay or kit | SimpleChIP Plus Sonication Chromatin IP Kit | CST | Cat# 56383S | |
| Chemical compound, drug | MG-132 | MCE | Cat# HY-13259 | |
| Chemical compound, drug | Bafilomycin A1 | MCE | Cat# HY-100558 | |

## Zebrafish maintenance and mutant generation

Zebrafish were raised, bred, and staged according to standard protocols (*Kimmel et al., 1995*). The following strains were used: AB, *Tg(mpeg1.1:eGFP)* (*Ellett et al., 2011*). For CRISPR9-mediated *zbtb14* knockout zebrafish generation, guide RNA (gRNA) targeting exon1 of *zbtb14* was designed using an online tool ZiFiT Targeter software (http://zifit.partners.org/ZiFiT), which was synthesized by cloning the annealed oligonucleotides into the gRNA transcription vector. Cas9 mRNA and gRNA were co-injected into one-cell stage zebrafish embryos. The injected F0 founder embryos were raised to adulthood and then outcrossed with wild type zebrafish. F1 embryos carrying potential indel mutations were raised to adulthood. Then, PCR amplification and sequencing were performed on genomic DNA isolated from tail clips of F1 zebrafish to identify mutants.

## Whole-mount in situ hybridization

Digoxigenin-labeled RNA probes were transcribed with T7, T3, or SP6 polymerase (Ambion, Life Technologies, Carlsbad, CA). WISH was performed as described previously (*Thisse and Thisse, 2008*). The probes labeled by digoxigenin were detected using alkaline phosphatase coupled anti-digoxigenin Fab fragment antibody (Roche, Basel, Switzerland) with 5-bromo-4-chloro-3-indolyl-phosphate nitro blue tetrazolium staining (Vector Laboratories, Burlingame, CA). Ten to 30 embryos were used for each probe. The positive signals were counted under a microscope, and the mean value was obtained from the counts of all of the embryos in the same group.

## Neutral red staining

Zebrafish larvae were collected at indicated time and soaked in Neutral Red (2.5 mg/ml, Sigma-Aldrich, St Louis, MO) overnight at 28.5°C. Staining was then observed under a microscope.

## Cell collection and FACS analysis

Cell collection and FACS analysis were performed as described (*Traver et al., 2003*). Wild type *Tg(mpeg1.1:eGFP)* and *zbtb14^{-/-}//Tg(mpeg1.1:eGFP)* larvae were dissociated into single cells using

0.05% trypsin (Sigma-Aldrich, St Louis, MO) as previously described (*Yan et al., 2015*). These disso-
ciated cells were passed through a 40 µm mesh, centrifuged at 450× *g*, and suspended in 5% FBS/
PBS before addition of propidium iodide to a final concentration of 1 µg/ml for exclusion of dead
cells. Wild type zebrafish (without GFP) were used as blank to determine the background values in
GFP controls. The GFP⁺ cells of each group were collected from a total of ~1000 larvae using a FACS
Vantage flow cytometer (Beckton Dickenson) (~300 larvae once, performed three times). For the
WKM samples, FACS analysis was based on forward and side scatter characteristics, propidium iodide
exclusion, and GFP fluorescence. The GFP⁺ cells in the myeloid gate was enriched from WKM samples
of wild type *Tg(mpeg1.1:eGFP)* and *zbtb14⁻ᐟ⁻//Tg(mpeg1.1:eGFP)* zebrafish (4-month-old, each time
one male and one female were used in the wild-type and mutant groups).

## pH3 staining and TUNEL assay

*Tg(mpeg1.1:eGFP)* and *zbtb14⁻ᐟ⁻//Tg(mpeg1.1:eGFP)* larvae were collected at 48 hpf and fixed in
4% paraformaldehyde. The fixed larvae were incubated with primary rabbit anti-phospho-histone H3
(pH3; Upstate Biotechnology) and goat anti-GFP (Abcam) antibodies according to the manufacturer's
protocol and subsequently stained with Alexa Fluor-647 anti-rabbit and Alexa Fluor-488 anti-goat
secondary antibodies (Invitrogen). TUNEL assays were performed using the In Situ Cell Death Detec-
tion Kit and TMR Red (Roche Diagnostics) according to the manufacturer's recommendations. Images
were taken using Olympus FV 1000 confocal microscopy equipped with the FV10-ASW version 3
software.

## MGG staining

FACS sorted cells were centrifuged by cytospin onto slides and stained with MGG (Sigma-Aldrich,
May-Grünwald solution, 63950 and Giemsa solution, 32884) following the manufacturer's instruc-
tions. Immature, intermediate, and mature monocytes/macrophages were counted based on their
morphology.

## RNA-seq and RT-qPCR

At 48 hpf, GFP positive cells were isolated from either wild type *Tg(mpeg1.1:eGFP)* or *zbtb14⁻ᐟ⁻//T-
g(mpeg1.1:eGFP)* larvae by FACS. mRNA was extracted from sorted cells using RNeasy Micro (Qiagen,
Manchester, UK) and mRNA libraries were constructed using NEBNext Ultra RNA Library Prep Kit for
52 Illumina and sequenced under Illumina HiSeq X Ten with pair end 150 bp (PE150).

The qPCR was carried out with SYBR Green Real-time PCR Master Mix (TOYOBO, Osaka, Japan)
with ABI 7900HT real-time PCR machine and analyzed with Prism software. *β-Actin* was served as
the internal control. The primers used are listed in *Table 1*. Each time a different batch of samples
was used. The expression levels of each interested gene were normalized to internal control *β-actin*
by RT-qPCR and compared with wild type group which was set to 1.0. RT-qPCR was performed with
gene-specific primers and gene expression levels were analyzed by comparative CT method.

## Plasmid construction

Zebrafish *zbtb14* gene and its serial mutants were cloned into PCS2⁺ vector. For the luciferase reporter,
the –0.6 kb promoter of zebrafish *pu.1* gene and the –1.1 kb promoter of zebrafish *zbtb14* gene were
cloned into the PGL3 basic vector (Promega, Madison, WI). Tol2-plasmid was constructed by insertion
of *pu.1* DN under *mpeg1.1* promoter (2 kb). Transgene was transiently expressed by co-injecting
80 pg of Tol2-plasmid and 120 pg of Tol2 transpose mRNA at one-cell stage. Primers used were listed
in *Table 1*.

## Morpholino and mRNA synthesis for microinjection

Zebrafish *zbtb14* (5'-ACTTCACAGTTTCGGACATACTGGA-3'), *pu.1* (5'-AATAACTGATACAAACTCAC
CGTTC-3') targeting the transcriptional initiation ATG of *zbtb14*, *pu.1* was designed and purchased
from Gene Tools. Full-length capped mRNA samples were all synthesized from linearized plasmids
using the mMessage mMachine SP6 kit (Invitrogen, Thermo Fisher Scientific, Waltham, MA). Micro-
injection concentration of mRNA was between 50 and 200 ng/µl and 2 nl of mRNA was injected at
one-cell stage embryos. All injections were performed with a Harvard Apparatus microinjector.

**Table 1.** Primers for plasmid generation, luciferase assays, qPCR, and ChIP-qPCR.

|  | Primers for plasmid generation |
| --- | --- |
| *zf zbtb14* | Forward 5'-CCGGAATTCTCCGAAACTGTGAAGTATGTG-3' |
|  | Reverse 5'-CCGCTCGAG TTATGAGCAGGCGATGGACTC-3' |
| *zf zbtb14$^{K40R}$* | Forward 5'-GTGGAGGATGTGAGGTTCAGGGCGCAT-3' |
|  | Reverse 5'ATGCGCCCTGAACCTCACATCCTCCAC-3' |
| *zf zbtb14$^{K259R}$* | Forward 5'-GCCACCGCTGACATGAGGTTTGAGTATCTGCTG-3' |
|  | Reverse 5'-CAGCAGATACTCAAACCTCATGTCAGCGGTGGC-3' |
| *tol2-mpeg1.1-pu.1 DBD* | Forward 5'-CGGGGTACCATGAATTCGCTTGTATCAGTTCCTGC-3' |
|  | Reverse 5'-CCGCAATTGTTAGAGAACCTCTCCACTGAACTGG-3' |
| *hs ZBTB14* | Forward 5'-CCGGAATTCATGGAGTTTTTCATCAGTATG-3' |
|  | Reverse 5'-CCGCTCGAGCTAGCTACAGGCTATCGTCTC-3' |
| *hs ZBTB14$^{S8F}$* | Forward 5'-GGAGACGTCAAAGTAAGAAAGGAGAGAGACCCTTGA-3' |
|  | Reverse 5'-TCAAGGGTCTCTCTCCTTTCTTACTTTGACGTCTCC-3' |
| *hs ZBTB14$^{S8A}$* | Forward 5'-TTCATCAGTATGGCTGAAACCATTAAA-3' |
|  | Reverse 5'-TTTAATGGTTTCAGCCATACTGATGAA-3' |
| *hs ZBTB14$^{S8D}$* | Forward 5'-TTCATCAGTATGGATGAAACCATTAAA-3' |
|  | Reverse 5'-TTTAATGGTTTCATCCATACTGATGAA-3' |
| *hs ZBTB14$^{S8W}$* | Forward 5'-TTTTTCATCAGTATGTGGGAAACCATTAAATAT-3' |
|  | Reverse 5'-ATATTTAATGGTTTCCCACATACTGATGAAAAA-3' |
| *hs ZBTB14$^{S8Y}$* | Forward 5'-CCGGAATTCATGGAGTTTTTCATCAGTATGTAT-3' |
|  | Reverse 5'-CCGCTCGAGCTAGCTACAGGCTATCGTCTCCAG-3' |
|  | **Primers for luciferase assays** |
| *zf zbtb14-Promoter* | Forward 5'-CGGGGTACCCATCAGTTGTATCTTAGGTACAG-3' |
|  | Reverse 5'-CCGCTCGAGTGGACTCCTCATGTTTGCTCT-3' |
| *zf pu.1-Promoter* | Forward 5'-CGGGGTACCACTAGTACACCTAAATTTATG-3' |
|  | Reverse 5'-CCGCTCGAGATTTGGCAGACCAACAACTGC-3' |
|  | **Primers for quantitative PCR** |
| *zf β-actin* | Forward 5'-TGCTGTTTTCCCCTCCATTG-3' |
|  | Reverse 5'-TTCTGTCCCATGCCAACCA-3' |
| *zf zbtb14* | Forward 5'-CTCGTGTGTTCGGCAGTAATTG-3' |
|  | Reverse 5'-CTGGAGCGAAATCCTTACTGG-3' |
| *zf pu.1* | Forward 5'-TCCCAGCAGTCGTAGTCCTC-3' |
|  | Reverse 5'-CCATTTCGCAGAAGGTCAA-3' |
| *zf c-myc* | Forward 5'-CAATTCTGGAACGGCATTCG-3' |
|  | Reverse 5'-GAAGTAGAAGTAGGGCTGGATG –3' |
| *hs β-ACTIN* | Forward 5'-CCAACCGCGAGAAGATGA-3' |
|  | Reverse 5'-CCAGAGGCGTACAGGGATAG-3' |
| *hs ZBTB14* | Forward 5'-CAGGATATGGGTCTGCAGGA-3' |
|  | Reverse 5'-TCTTAATGCCTTGAACGCCA-3' |
|  | **Primers for ChIP-qPCR** |

*Table 1 continued on next page*

*Table 1 continued*

| | Primers for plasmid generation |
|---|---|
| Negative control primer | Forward 5'-GCTGAAATTTTGTAGTCTGTC-3' |
| | Reverse 5'-ATAAGATTTTAGTCATCAAAC-3' |
| Positive primer | Forward 5'-ACTAGTACACCTAAATTTATG-3' |
| | Reverse 5'-GACAGACTACAAAATTTCAGC-3' |

## Cell culture and luciferase reporter assay

HEK293T was obtained from ATCC. Identity has been authenticated by STR profiling, and the cell line tested negative for mycoplasma. HEK293T cells were maintained in DMEM (Gibco, Life Technologies, Carlsbad, CA) with 10% fetal bovine serum (Gibco, Life Technologies, Carlsbad, CA). Plasmid transfection was carried out with Effectene Transfection Reagent (Qiagen, Manchester, UK) according to the manufacturer's instruction. For the luciferase reporter assay, cells were harvested 48 hr after transfection and analyzed using the Dual Luciferase Reporter Assay Kit (Promega, Maddison, WI), according to the manufacturer's protocols. Primers used were listed in *Table 1*.

## Chromatin immunoprecipitation PCR

For ChIP analysis, GFP and GFP-Zbtb14 expressing larvae were harvested at 48 hpf for brief fixation. Cross-linked chromatin was immunoprecipitated with anti-GFP antibody according to the procedure described (*Hart et al., 2007*). The resultant immunoprecipitated samples were subjected to quantitative PCR using primer pairs (*Table 1*).

## Statistical analysis

Data were analyzed by SPSS software (version 20) using two-tailed Student's t test for comparisons between two groups and one-way analysis of variance (ANOVA) among multiple groups. Differences were considered significant at $p<0.05$. Data are expressed as mean ± standard error of the mean (SEM).

# Acknowledgements

The authors are grateful to Y Chen and Y Jin (both from Shanghai Jiao Tong University School of Medicine, Shanghai, China) for technical support. We thank Dr X Jiao (from the Department of Cell Biology and Neuroscience, Rutgers University, Piscataway, NJ) for his critical manuscript reading.

# Additional information

### Funding

| Funder | Grant reference number | Author |
|---|---|---|
| National Natural Science Foundation of China | NO.32171097 | Jun Zhou |

The funders had no role in study design, data collection and interpretation, or the decision to submit the work for publication.

### Author contributions

Yun Deng, Data curation, Formal analysis, Methodology; Haihong Wang, Data curation, Methodology; Xiaohui Liu, Data curation; Hao Yuan, Formal analysis; Jin Xu, Hugues de Thé, Resources; Jun Zhou, Resources, Supervision, Funding acquisition, Validation, Writing – original draft, Project administration; Jun Zhu, Resources, Supervision, Validation, Writing – original draft, Project administration

### Author ORCIDs

Jin Xu http://orcid.org/0000-0002-6840-1359

Jun Zhou  http://orcid.org/0000-0003-0472-3188
Jun Zhu  http://orcid.org/0000-0002-7983-3130

### Ethics

The study was approved by the Ethics Committee of Rui Jin Hospital Affiliated to Shanghai Jiao Tong University School of Medicine. Zebrafish experimental procedures were conducted in accordance with the protocols approved by the Institutional Animal Care and Use Committee (IACUC) of Shanghai Jiao Tong University (2020-3#).

### Decision letter and Author response

Decision letter https://doi.org/10.7554/eLife.80760.sa1
Author response https://doi.org/10.7554/eLife.80760.sa2

## Additional files

### Supplementary files
- MDAR checklist
- Source data 1. Original blots.

### Data availability

RNA sequencing dataset generated in this study was deposited with Dryad-https://doi.org/10.5061/dryad.9cnp5hqms.

The following dataset was generated:

| Author(s) | Year | Dataset title | Dataset URL | Database and Identifier |
|-----------|------|---------------|-------------|-------------------------|
| Deng Y, Wang H, Liu X, Yuan H, Xu J, de Thé H, Zhou J, Zhu J | 2022 | RNA SEQ | https://dx.doi.org/10.5061/dryad.9cnp5hqms | Dryad Digital Repository, 10.5061/dryad.9cnp5hqms |

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
