## [Editor Report]

This manuscript by Deng et al. is a valuable evaluation of zbtb14 and its role in normal myelopoiesis. The authors provided convincing data supporting the role played by zbtb14 in monocyte and macrophage development and its regulation involving the modulation of PU.1 expression. The finding that a mutation in ZBTB14 exists in AML patients also implies how important this gene product is in normal human myelopoiesis.

---

## [Decision Letter]

**Decision letter after peer review:**

Thank you for submitting your article "Zbtb14 regulates monocyte and macrophage development through inhibiting pu.1 expression in zebrafish" for consideration by *eLife*. Your article has been reviewed by 2 peer reviewers, and the evaluation has been overseen by a Reviewing Editor and Satyajit Rath as the Senior Editor. The reviewers have opted to remain anonymous.

Essential revisions:

Below are the two major points that we think will be important to address to improve the manuscript. More details on these points can be found in the reviewer comments.

1) Provide more evidence on how pu.1 overexpression mechanistically causes Mac expansion? Deeper analysis of the RNASeq data might help.

2) Better differentiate between expanded mature macs and progenitors.

*Reviewer #1 (Recommendations for the authors):*

Although the findings are interesting, but several questions need to be addressed to justify its publication.

1. In figure 2, the authors show zbtb14 mutants exhibited abnormal expansion of macrophages in both embryonic and adulthood stages. However, it remains unclear whether this phenotype is due to the enhanced proliferation of macrophages or myeloid progenitors. Cell proliferation assays (like BrdU staining or anti-PH3 immunostaining) in mpeg1.1+ or pu.1+ cells would be necessary to determine the proliferative ability of embryonic macrophages or myeloid progenitors. In parallel, the FACS-based WKM analysis or May-Grunwald-Giemsa (MGG) staining can also be performed to examine whether there is an increased progenitors/myeloblasts in zbtb14 mutants.

2. In figure 3, despite the evidence of the suppression role of Zbtb14 in the regulation of pu.1 expression, whether the macrophage expansion in zbtb14 mutants is indeed caused by the increased pu.1 expression remains unclear. Although macrophage-specific knockdown of pu.1 activity assay in zbtb14 mutants by injecting mpeg1.1:pu.1-DBD construct could rescue the mutant phenotype, it may not be faithful, as Pu.1 activity may be over-suppressed, thus leading to the impairment of macrophage lineage development instead of restraining the macrophage expansion. Likewise, the change of macrophage numbers between zbtb14+/+pu.1△371/△371 and zbtb14-/-pu.1△371/△371 embryos could also be caused by the loss of pu.1 function, which impairs the early development of macrophages. Given these potential problems, it would be necessary for authors to test: (1) whether there is any defect in myeloid lineage development in the rescued fish; (2) whether reduced pu.1 expression to normal level would be able to correct the mutant defect in zbtb14 mutants or whether overexpressing pu.1 in macrophages in late stage of WT fish can indeed expand macrophage population without interfering myeloid lineage development.

3. It would be necessary to show the expression pattern and level of zbtb14, especially in different hematopoietic lineages at different developmental stages, which may provide additional evidence supporting the role of zbtb14 in the macrophage expansion.

4. In figure 3 and lines 147-148, the authors performed RNA-seq of mpeg1.1+ macrophages in zbtb14 mutants and siblings and identified an upregulation of pu.1 expression in zbtb14 mutants. Please show the corresponding pu.1 FPKM/TPM values for the comparison. Besides, a more detailed analysis, such as the differentially expressed genes (DEGs) and the biological implications of DEGs, of the RNA-seq data is needed to provide molecular insights into the role of zbtb14 in macrophage development.

5. Given the pro-proliferation role of c-myc and the elevated expression level of this gene in zbtb14-/- macrophages, it would be interesting to test whether c-myc plays a role in the macrophage expansion in the zbtb14-/- embryos.

*Reviewer #2 (Recommendations for the authors):*

Overall, I think the manuscript is very strong. The data is very solid, and the story is well written. I had only one question:

1. In figures (say, Figure 1), could the gene being evaluated be written in each box? For instance, C and C' are both evaluations of mfap4 (I assume), but that is not clear from the figure. Is there a better way to show those data?

---

## [Author Response]

Reviewer #1 (Recommendations for the authors):Although the findings are interesting, but several questions need to be addressed to justify its publication.1. In figure 2, the authors show zbtb14 mutants exhibited abnormal expansion of macrophages in both embryonic and adulthood stages. However, it remains unclear whether this phenotype is due to the enhanced proliferation of macrophages or myeloid progenitors. Cell proliferation assays (like BrdU staining or anti-PH3 immunostaining) in mpeg1.1+ or pu.1+ cells would be necessary to determine the proliferative ability of embryonic macrophages or myeloid progenitors. In parallel, the FACS-based WKM analysis or May-Grunwald-Giemsa (MGG) staining can also be performed to examine whether there is an increased progenitors/myeloblasts in zbtb14 mutants.

Since *mpeg1.1* is a macrophage-specific marker (*Ellett, et al.*, *2011*), we used a *mpeg1.1* promoter-driven reporter line to monitor macrophage development in this study.

The aberrant expansion of macrophage population in *zbtb14*-defective mutants can result from either abnormalities in proliferation or apoptosis rate. As suggested by the reviewer, to distinguish between the two possibilities, antiphosphohistone H3 (pH3) immunostaining and terminal deoxynucleotidyl transferase dUTP nick end labeling (TUNEL) assays were carried out to evaluate the proliferation and apoptosis status of macrophages, respectively.

A significant increase of pH3^+^GFP^+^ double-positive cells were observed in *zbtb14^-/-^*//Tg(*mpeg1.1:*eGFP) larvae compared to those in controls (Figure 3A-C’, D). Nevertheless, the results from TUNEL assay could not show discernible differences in the percentage of TUNEL^+^GFP^+^ cells in *zbtb14^-/-^*//Tg(*mpeg1.1:*eGFP) larvae compared to the percentage in controls (Figure 3E-G’, H). These data suggest that the abnormal expansion of macrophages is due to enhanced proliferation.

As the amount of the *mpeg1.1^+^* cells isolated from the zebrafish larvae was not enough to be analyzed, the *mpeg1.1*^+^ cells were enriched from the myeloid fraction of WKM in adult zebrafish and subjected to May-Grünwald-Giemsa (MGG) staining. According to their morphology, the monocytes/macrophages were classified into three populations: proliferative immature progenitors (Figure 3I), intermediate (Figure 3II), and non-proliferative mature cells (Figure 3III). The results revealed that the proportion of immature progenitors (with a round-shape morphology and a higher nuclear/cytoplasm ratio) was obviously increased in *zbtb14* mutants compared with the siblings (Figure 3I, J). By contrast, the proportion of fully differentiated macrophages was decreased in the mutants (Figure 3I, J).

Based on these observations, we can draw the conclusion that the abnormal expansion of macrophages is mainly due to excessive monocyte/macrophage progenitor proliferation in *zbtb14* mutants. These results have been added in the revised manuscript.

2. In figure 3, despite the evidence of the suppression role of Zbtb14 in the regulation of pu.1 expression, whether the macrophage expansion in zbtb14 mutants is indeed caused by the increased pu.1 expression remains unclear. Although macrophage-specific knockdown of pu.1 activity assay in zbtb14 mutants by injecting mpeg1.1:pu.1-DBD construct could rescue the mutant phenotype, it may not be faithful, as Pu.1 activity may be over-suppressed, thus leading to the impairment of macrophage lineage development instead of restraining the macrophage expansion. Likewise, the change of macrophage numbers between zbtb14+/+pu.1∆371/∆371 and zbtb14-/-pu.1△371/△371 embryos could also be caused by the loss of pu.1 function, which impairs the early development of macrophages. Given these potential problems, it would be necessary for authors to test: (1) whether there is any defect in myeloid lineage development in the rescued fish; (2) whether reduced pu.1 expression to normal level would be able to correct the mutant defect in zbtb14 mutants or whether overexpressing pu.1 in macrophages in late stage of WT fish can indeed expand macrophage population without interfering myeloid lineage development.

Actually we also performed the WISH assays in *zbtb14^-/-^* mutants injected with the *mpeg1.1:pu.1-DBD* construct. The results showed that while the aberrant macrophage expansion could be effectively rescued, the development of neutrophil lineage still kept normal (Figure 4—figure supplement 2A-A’’, B). These findings suggest the dominant negative Pu.1 has no impact on neutrophil-macrophage progenitor (NMP) development.

Consistently, the injection of macrophage-specific full-length *pu.1* (TOL2-*mpeg1.1*:*pu.1* construct) in wild type embryos could induce an expansion of macrophage population without interfering neutrophil lineage development (Figure 4—figure supplement 2C-D’, E). These observations further suggest the role of *zbtb14* is macrophage-lineage-specific. These results have been added in the revised manuscript.

3. It would be necessary to show the expression pattern and level of zbtb14, especially in different hematopoietic lineages at different developmental stages, which may provide additional evidence supporting the role of zbtb14 in the macrophage expansion.

As suggested by the reviewer, the GFP-positive cells were isolated from wild type Tg(*mpeg1.1:*eGFP) and Tg(*mpx:*eGFP) lines at 2 dpf and 5 dpf, respectively. RT-qPCR analyses revealed that the expression level of *zbtb14* in macrophages was much higher than that in neutrophils at different developmental stages (see below), which may provide additional evidence supporting the role of *zbtb14* in the macrophage expansion. This result has been added in Figure 4—figure supplement 3.

4. In figure 3 and lines 147-148, the authors performed RNA-seq of mpeg1.1+ macrophages in zbtb14 mutants and siblings and identified an upregulation of pu.1 expression in zbtb14 mutants. Please show the corresponding pu.1 FPKM/TPM values for the comparison. Besides, a more detailed analysis, such as the differentially expressed genes (DEGs) and the biological implications of DEGs, of the RNA-seq data is needed to provide molecular insights into the role of zbtb14 in macrophage development.

The corresponding *pu.1* (also known as *spi1b*) FPKM values (duplicate) for the comparison are shown in Author response image 1:

**Author response image 1. sa2fig1:** wt: *Tg(mpeg1.1:eGFP)*; homo: *zbtb14^-/-^*//*Tg(mpeg1.1:eGFP)*.

The differentially expressed genes (DEGs) analysis of the RNA-seq data indicated that multiple important regulators involving monocyte/macrophage development such as *pu.1*, *csf1ra*, *csf1b*, *il34*, *klf4*, *mafba*, *mafbb*, *irf8*, and *c-myc* (also known as *myca*) were upregulated in mutant macrophages (see Figure 4—figure supplement 1).

PU.1, the most important transcription factor in monocyte/macrophage development, can promote macrophage proliferation (*Celada, et al.*, *1996*). Besides, M-CSFR signaling is also required for the proliferation of monocytes/macrophages. M-CSFR-deficient mice show severely reduced numbers of most tissue-resident macrophages and monocytes (*Dai, et al.*, *2002*). The expression level of *pu.1*, *csf1ra* (orthologous to human *MCSF-R*) and its two ligands *csf1b* (orthologous to human *MCSF*) and *il34* are all increased in the *zbtb14* mutant macrophages, which would be the reason underlying the abnormal macrophage expansion. These results have been added in the revised manuscript.

5. Given the pro-proliferation role of c-myc and the elevated expression level of this gene in zbtb14-/- macrophages, it would be interesting to test whether c-myc plays a role in the macrophage expansion in the zbtb14-/- embryos.

A *mpeg1.1* promoter driven dominant-negative Tol2-C-myc DBD construct was injected in *zbtb14^-/-^*//Tg(*mpeg1.1:*eGFP) embryos according to the suggestion. A slight amelioration of macrophage expansion could be observed (see Author response image 2). Yet, the rescue effect of C-myc DBD is much weaker than that of Pu.1 DBD. Such findings suggest although *c-myc* may also play a role in the macrophage expansion, its contribution is less important than *pu.1*.

References:Celada A, Borras FE, Soler C, Lloberas J, Klemsz M, van Beveren C, McKercher S, Maki RA. 1996. The transcription factor PU.1 is involved in macrophage proliferation. *J EXP MED* 184:61-69. doi:10.1084/jem.184.1.61

Dai XM, Ryan GR, Hapel AJ, Dominguez MG, Russell RG, Kapp S, Sylvestre V, Stanley ER. 2002. Targeted disruption of the mouse colony-stimulating factor 1 receptor gene results in osteopetrosis, mononuclear phagocyte deficiency, increased primitive progenitor cell frequencies, and reproductive defects. *BLOOD* 99:111-120. doi:10.1182/blood.v99.1.111

Ellett F, Pase L, Hayman JW, Andrianopoulos A, Lieschke GJ. 2011. mpeg1 promoter transgenes direct macrophage-lineage expression in zebrafish. *BLOOD* 117:e49-e56. doi:10.1182/blood-2010-10-314120

Reviewer #2 (Recommendations for the authors):Overall, I think the manuscript is very strong. The data is very solid, and the story is well written. I had only one question:1. In figures (say, Figure 1), could the gene being evaluated be written in each box? For instance, C and C' are both evaluations of mfap4 (I assume), but that is not clear from the figure. Is there a better way to show those data?

We are grateful for the positive comments of our manuscript, and the figure legends have been modified as suggested by reviewer #2.